# Barriers for Sports and Exercise Participation and Corresponding Barrier Management in Cystic Fibrosis

**DOI:** 10.3390/ijerph192013150

**Published:** 2022-10-13

**Authors:** Stefanie Dillenhoefer, Florian Stehling, Matthias Welsner, Anne Schlegtendal, Sivagurunathan Sutharsan, Margarete Olivier, Christian Taube, Uwe Mellies, Cordula Koerner-Rettberg, Folke Brinkmann, Wolfgang Gruber

**Affiliations:** 1Department of Pediatric Pulmonology, Cystic Fibrosis Center, University Children’s Hospital of Ruhr University Bochum at St. Josef-Hospital, 44791 Bochum, Germany; 2Pediatric Pulmonology and Sleep Medicine, Cystic Fibrosis Center, Children’s Hospital, University of Duisburg-Essen, 45117 Essen, Germany; 3Department of Pulmonary Medicine, Adult Cystic Fibrosis Center, University Hospital Essen-Ruhrlandklinik, University of Duisburg-Essen, 45239 Essen, Germany; 4Children’s Hospital Marienhospital Wesel, 46483 Wesel, Germany; 5Institute of Human Nutrition and Food Science, Christian Albrechts University Kiel, 24118 Kiel, Germany

**Keywords:** cystic fibrosis, physical activity, barrier, barrier management, counter strategy, exercise program

## Abstract

Background: Nowadays physical activity (PA)/exercise is an important component of cystic fibrosis (CF) therapy. The aim of the study was to assess the barriers to PA and the barrier management and to explore the effect of supervision on the barriers and barrier management during an exercise program. Methods: In total, 88 people with CF (pwCF) of the ages 6 to 50 years old (mean 24.2 ± 7.9 yrs) participated in the partially supervised 12-month exercise program and filled in a structured and validated questionnaire about barriers to sports and barrier management at baseline. Additionally, 23 pwCF filled in the questionnaire after 6 months and 12 months. The items were clustered into physical and psychosocial barriers and into preventive counter strategies and situational counter strategies and analyzed at baseline and over time. Results: Physical barriers were more relevant than psychosocial barriers and no trend could be seen in the situational and preventive counter strategies. When divided in subgroups, the less active pwCF (<7500 steps/day), more active pwCF (>7500 steps/day), physical barriers, and psychosocial barriers showed no significant differences. However physical barriers showed a tendency to have a higher value in the less active group compared to the more active group (*p* > 0.05). Stratified by age or FEV1%pred between the subgroups, no differences could be seen regarding barriers and counter strategies. Conclusions: Physical barriers seemed to have a higher priority when it comes to not participating in PA/exercise. Supervision over 6 months during an exercise program did not show a beneficial effect on barriers and barrier management. Besides the motivational aspect of sport counselling, the volitional aspect seemed to be more important to incorporate more PA into daily life. Individual barriers and their concrete counter strategies should be discussed with the patient with CF. Sport counselling is needed permanently and should be part of the CF routine care.

## 1. Introduction

Cystic fibrosis (CF) is an autosomal recessive multi-organ disease caused by mutations in the cystic fibrosis transmembrane conductance regulator (CFTR) gene. This genetic disorder leads to the absence or reduction in the quantity of the encoded protein, which is an anion transporter located on epithelial surfaces. The result is a disturbed transport of chloride and bicarbonate ions, which are responsible for an abnormal viscous mucus and, therefore, multi-organ dysfunction [1]. Morbidity and mortality are mainly influenced by pulmonary involvement [2]. There is currently no known cure. Due to advances in care, the life expectancy of people with CF (pwCF) has increased significantly over the last decades [3]. However, therapy is time consuming, comprising a complex regime of pharmacological treatments, physiotherapy, airway clearance, high calorie diets, exercise, and physical activity [4].

In the last years, exercise and physical activity (PA) have become an important component of therapy of CF [5,6]. Positive effects on lung function FEV1 [6] and a slowdown of the annual rate of lung function decline were shown [6,7]. Long-term effects could be seen in improved peak oxygen uptake (VO2peak) and improved peak workload (Wpeak) [7]. Additionally, PA and exercise are associated with improved bone mineral density [8], a greater BMI [9], an improved probability of survival, and health-related quality of life [10].

The general recommendations of the German national guideline for PA for healthy children aged 6 to 11 years and adolescents aged 12 to 18 years are similar to the evidence-based recommendations for children and adolescents with CF and include 90 min of physical activity per day. They should spend at least 60 min of moderate-to-vigorous PA daily. Aerobic exercise should be performed for 30–60 min, 3 times per week or more, to improve their aerobic exercise capacity. Additionally, it is recommended that they do resistance training 2–3 times per week [11,12]. The PA recommendation for healthy adults and adults with CF is 150 min time of physical activity per week at a moderate intensity or 75 min/week of aerobic physical activity at a vigorous intensity. Additionally, resistance training is recommended 2–3 times per week [11,12]. However, studies have shown that most of the pwCF did not meet these recommendations due to barriers related to the burden of the daily routine care [13]. However, healthy children and adolescents often do not meet the recommendations [14], while only 20.5% of healthy women and 24.7% of healthy men in Germany meet the WHO recommendations [15].

A recently published systematic review in children, adolescents, and young adults with CF showed that aspects like perceptions of PA, the value for PA, social influences, competing priorities, fluctuating health, normality, control beliefs, coping strategies, and availability of facilities may serve as barriers to, as well as facilitators of, participation in PA [16]. Almost all studies included used interview techniques without an exercise program. Only one out of the seven studies combined an interview asking about barriers to and facilitators of PA participation with a 2-month exercise program and repeated the interview 6 months later. One of the primary motivators was parent–family participation. However, it could be seen that motivation declined and the novelty wore off for several (approximately half) parent-child dyads who planned to decrease or stop the exercise program after the study ended [16].

However, there is little knowledge about barriers to sports and counter strategies in CF in combination with an exercise program lasting more than 2 months and a possible change to barriers to sports and counter strategies over time.

In patients with non-specific chronic low back pain, a lack of time, lack of motivation, and lack of willingness to participate in PA were found to be main barriers to physical exercise [17]. In people with COPD participating in a pulmonary rehabilitation, besides information and educational aspects, other key facilitators included continuing support from health care professionals, continuing peer interaction, and opportunities to access PA maintenance groups following pulmonary rehabilitation. Anxiety and fear, restricted access to social support, and lack of positive feedback served as barriers and leads to a return to previous habits formed prior to pulmonary rehabilitation [18]. In children with disability, personal barriers (e.g., lack of skills, fear, lack of knowledge), social barriers (e.g., parental actions, negative societal attitudes to disability), environmental barriers (e.g., lack of transport, accessibility of facilities), and policy or program barriers (e.g., lack of appropriate physical activity programs, costs) seemed to be barriers to participation in regular sports and exercise. Facilitators included the child’s desire to be active, practising skills, involvement of peers, family support, accessible facilities, proximity of location, better opportunities, skilled staff, and information [19].

The aim of our study was to assess barriers to participation in pwCF and the management of these barriers using a validated questionnaire [20]. The second aim of the study was to examine the effects of a partially supervised long term exercise program on barriers and barrier management under supervision and after supervision stopped.

We assume that physical and psychological barriers play a more important role in less active pwCF than in more active pwCF and that physical training and supervision have a positive effect on barriers and barrier management in pwCF.

## 2. Materials and Methods

CFmobil was a partially supervised sport study/program performed from July 2014 to August 2018 at three regional CF centers (Ruhrlandklinik Essen, University Children’s Hospital Essen, and University Children’s Hospital Bochum) in Germany. The study was approved by the local ethics committees of the University Hospital Essen (14-6117-BO) and University Hospital Bochum (15-5314-BR) and listed in the Clinical Trials (NCT03518697). Written informed consent was provided by all participants.

Inclusion criteria for the participation were an age ≥ 6 years, confirmed diagnosis of CF by at least two pathologic sweat tests and/or by the presence of two CF mutations, willingness to participate in and to comply with the research project procedure, and written informed consent of patients and parents. Subjects having one or more of the following indications were excluded: severe pulmonary exacerbation, Cor pulmonale, musculoskeletal discomfort that makes a regular exercise training impossible, and untreated CF-related diabetes.

In total, n = 88 pwCF (38/51 m) aged 6–50 years (mean 24.2 ± 7.9 yrs) agreed to participate in the CFmobil exercise program. Of these 88 pwCF, n = 23 pwCF aged from 14 to 50 years old (mean 25.5 ± 8.9 yrs) completed the questionnaire at all testing times. (Figure 1)

### 2.1. Study Design

Anthropometry, lung function, physical fitness, and motor performance were assessed at time baseline (T1), after 6 months (T3), and after 12 months (T4) (Figure 2).

The individual exercise program was created based on the patient’s exercise test, age, environmental factors, and individual interests. The counselling included a great variety of physical activities, for example, trend sports, traditional sport activities, video games (Wii sports^®^, Microsoft Xbox^®^ with Kinect device), and games enhancing motor skills. An increase of their daily PA by at least 30–60 min/week in addition to school sports or other exercise activities were suggested to all pwCF, and the participants could borrow training equipment within the period of the study. In a training diary, all activities were recorded. During the first 6 months (from T1 to T3) of training, participants or, for younger children, their parents had close contact with the sports and exercise scientists via biweekly telephone calls and outpatient visits at intervals of 3 months, which are periods with intensive supervision. In the following 6 months (from T3 to T4), monitoring and supervision was stopped. The pwCF or their parents had the possibility to contact the exercise scientist or supervising physician as needed.

### 2.2. Testing

Anthropometry and lung function: Bodyweight was recorded using an electronic flat scale (seca 861, seca, Hamburg, Germany). Height was determined exactly to 0.1 cm using a telescopic measuring rod (seca 202, seca Hamburg, Germany). Body mass index (BMI) was calculated (kg/m^2^). FEV1 (forced expiratory volume in the first second) and forced vital capacity (FVC) were measured by standard spirometric techniques (JAEGER MasterScreen Body, Care Fusion, Hoechberg, Germany) according the ATS guidelines [21]. All subjects completed an incremental cycling test using the Godfrey protocol (ergoselect 100 p, ergoline GmbH, Bitz, Germany or ViaSprint 200, ergoline GmbH, Germany) to determine the peak exercise capacity, expressed as peak workload and peak heart rate [22]. Heart Rate was monitored continuously using a Heart Rate monitor and chest strap (polar A300, Polar, Kempele, Finland). Peak heart rate and peak workload were taken from the highest value in the last 30 sec before stopping the test. The percentage of the predicted values were determined from the reference equations of Godfrey for the workload and of Rowland for Heart Rate [22,23]. Habitual physical activity was determined using an activity monitor (wActiSleep-BT Monitor, Actigraph Corp., Pensacola, FL, USA) for 4 weeks before the intervention started, after three months, and at the end. Questionnaire: Participants filled in the validated questionnaire for self-evaluation to assess motivational and volitional aspects underlying participation in sport and exercise activities on the same day that the exercise testing was performed [20]. The questionnaire included 13 items to gather information on barriers to exercise and sport. The barrier items assessed using the questionnaire were grouped in psychosocial barriers and physical barriers [20]. Based on a 4-point Likert scale (no relevance, poor relevance, important, and very important) participants could answer as to what extent the statement prevents them from participating in exercise and sport. The assessment of the barrier management was taken using 14 questions, and answers were divided into preventive counter strategies and situational counter strategies. These strategies described potential solutions to surmount the individual’s barriers to participating in exercise and sport activities. As with the questions about the barrier management, the patients were able to make a selection based on a 4-point Likert-Scale (strongly disagree, somewhat, agree, and strongly agree). The answers are indicated by the numbers 1 to 4. The higher the value of the answer, the higher the consent.

### 2.3. Statistical Analysis

Descriptive statistics (mean, standard deviations) were calculated for all variables. The test items of the questionnaire were clustered into four groups: psychosocial barriers, physical barriers, situational counter strategies, and preventive counter strategies. In order to evaluate the data according to the severity of the disease, the participants were divided into two activity groups (more active pwCF with >7500 steps/day and less active pwCF with <7500 steps/day). Additionally, the participants were divided into pediatric and adult populations. Furthermore, the participants were stratified by FEV1pred% less than 40% and more than 40%. All data were checked for normal distribution. The data were evaluated using a Friedman One-Way Repeated Measure Analysis of Variance by Ranks. Differences between measurements were analyzed by a Wilcoxon signed-rank test. A Mann–Whitney test was used to compare the parameters between different activity groups. All statistical analyses were computed by SPSS version 22.0 (SPSS Inc., Chicago, IL, USA). A *p*-value of ≤0.05 was considered statistically significant.

## 3. Results

### 3.1. Anthropometric Characteristics at Baseline

Table 1 shows the anthropometric parameters, BMI, lung function, PA in steps per day, intensity of PA in different intensity levels, Pmax (peak workload), and results of barriers and counter strategies of the 88 participants (n = 88) at starting point T1, identifying the study cohort as the representative CF patient cohort. Physical activity was expressed in the metabolic equivalent of task (MET). One MET is the amount of energy used while sitting quiet.

### 3.2. Anthropometric Characteristics in Less and More Active Participants

Table 2 shows the anthropometric parameters, lung function, and activity parameters in the two activity groups (less active pwCF < 7500 steps/day and more active pwCF > 7500 steps/day). The two groups did not differ in age, but BMI was significantly higher in the more active groups with BMI being normal, whereas patients in the less active group were slightly underweight. (*p* ≤ 0.05). Also, ppFEV1 and ppFVC were significantly higher in the more active group of pwCF (*p* ≤ 0.05). The more active pwCF showed a significantly better performance in terms of maximal workload on exercise testing (*p* ≤ 0.05). Regarding the intensity level of the PA, the less active group spent significantly more time in sedentary intensity and less time in moderate to vigorous intensity (*p* > 0.05).

Regarding physical barriers like illness, pain, or injury, no significant difference could be found between the two groups. However, at timepoint T1, a small tendency towards a higher relevancy of physical barriers in the less active group could be described compared to the more active group without gaining significance (*p* > 0.05). Neither the psychosocial barriers nor the situational and preventive counter strategies showed differences between the more active and less active group. 

Table 3 shows the participants divided into pediatric and adult populations. Physical activity, measured in steps per day, in children is higher than in the adult population, without significance. Additionally, no significant differences could be observed in physical barriers, psychosocial barriers, or situational and preventive counter strategies between the two groups. However, a small tendency towards more importance of physical barriers in the adult group could be seen, without significance.

Table 4 describes the subgroups stratified by FEV1%pred in a subgroup with a FEV1%pred of less than 40% and a subgroup with a FEV1%pred of more than 40%. The pwCF with a FEV1%pred of less than 40% were less physically active with significantly lower steps per day compared to the subgroup with a FEV1%pred of more than 40% (*p* < 0.005). No significant changes could be shown concerning physical barriers, psychosocial barriers, or situational and preventive counter strategies between the groups.

For 23 patients, complete records (including questionnaires of barriers and barrier management) were available at all time points. In both groups, i.e., the more active and the less active groups, the ppFEV1 and bodyweight (kg) remained stable over the entire time (*p* > 0.05). In contrast, the ppFVC in the more active pwCF showed a significant decline from T1 to T4 (*p* < 0.05) while maximal workload and ppFEV1 did not change. The group of less active patients showed a significantly higher gain in performance in terms of maximal workload from T1 to T4 (*p* < 0.05) (Table 5), as well as an increase in step count over time; the latter, however, did not show significance. The time for the sedentary intensity level decreased significantly when comparing T1 and T4 (*p* ≤ 0.05). Correspondingly, the time in the higher intensity level increased non-significantly over time in less active pwCF.

Looking at the physical barriers over time, no significant changes (*p* > 0.05) were seen between the timepoints although we saw a trend in which the physical barriers lost their importance over time in the less active group. Long-term observation showed no significant change in psychosocial barriers in both groups.

In both groups, no significant differences or changes over time were found for the preventive and situational counter strategies after the 6-month supervised exercise program nor when supervision was stopped.

## 4. Discussion

To our knowledge, this is the first study investigating barriers to and barrier management of exercise and sports in pwCF using a validated questionnaire with motivational and volitional aspects. Another new aspect is the combination of a partially supervised exercise program and reevaluation at three different time points. Using a previously validated questionnaire, predefined items were asked in a structured way and answers were compared over time [20]. In the context of the sports counseling of patients, it is useful to distinguish between physical barriers (e.g., illness, injury, or pain) and psychological barriers (e.g., tiredness, stress, not being in the mood for sports, weather, and having a cozy home). Physical and psychological barriers both separately and in combination may have different effects on the participation in PA and sports. However, in the present study, no significant differences were found in situational physical or situational psychosocial barriers. Situational physical barriers appeared to be a more important reason than situational psychosocial barriers for not participating in sport and exercise (Table 1).

By splitting the cohort into a more active (>7500 steps per day) and a less active group, (<7500 steps per day) we observed a tendency of physical barriers in the less active group to have higher relevance. This may be explained by a higher severity of lung disease and a lower ppFEV1 in the less active group. In line with this, it was shown that the pwCF with a FEV1%pred < 40% were significantly less physically active compared to the pwCF with a FEV1%pred > 40% (Table 4). Higher physical barriers might be related to the fear of pwCF of physical disadvantages of PA (e.g., dyspnea, exercise-induced bronchoconstriction) [24]. However, the overall safety of exercise or PA for pwCF was shown by Ruf et al. who investigated adverse reactions during exercise testing and adverse reactions associated with exercise (e.g., pneumothorax). Although a variety of adverse reactions in the pwCF (e.g., pneumothorax, cardiac arrhythmia, injury, or hypoglycaemia) might occur, the real incidence is low (<1% each) for both in-hospital physical training as well as exercise in daily life [25]. For this reason, education about the good safety profile and potential risks in sports activities in CF is mandatory. Individual limitations can be detected by an exercise test. PwCF should learn that a training effect can only be achieved with a moderate intensity of PA, not with PA of light intensity [12].

The causal relationship between lower levels of PA and ppFEV1 observed here is debatable. Whether the ppFEV1 is lower in less active people because they exercise less or whether the less active people exercise less because of the poor ppFEV1 remains open. The pwCF in the less active group clearly had more advanced lung disease and, therefore, may have had a limitation in PA. A strong association between a physically inactive lifestyle and CF disease progression has been described previously [7,26].

Divided in subgroups by age, no significant differences in barriers or counter strategies could be observed; there was a slight tendency of physical barriers to be more important in the adult group but this was without significance. We assume that a larger number of children with CF included within the study could possibly show a clearer difference. 

Looking at the different test times, the situational physical and situational psychosocial barriers in the less and more active groups did not change between the three different timepoints and was independent of supervision. Over time, physical barriers became less important in the less active group with pwCF. Whether this trend is attributable to personal contact with the sports therapists and the knowledge transfer about the general effects and safety of exercise, or rather, to an increased personal fitness level or subjectively perceived ease to perform exercise remains open. In this respect, it is of note that the maximum workload (but not ppFEV1) increased significantly in the less active group. This increase corresponds to the observed responsiveness to training and higher improvement in subjects with a lower initial fitness level during a 6-week exercise program [27].

Regarding the results of barrier management, there was no significant change over time in both groups. The supervision did not affect preventive and situational counter strategies (Table 5). Steps per day increased in both groups non-significantly during the supervised period from T1 to T3. In the less active group, the increase of steps per day continued to increase after supervision was stopped. However, in the more active group the steps per day dropped again from T3 to T4 without supervision (*p* > 0.05). The supervision was a facilitator to motivate the pwCF to improve daily PA. However, not all study participants met the German national guideline recommendation and the evidence-based recommendations for PA for pwCF neither before, during, nor after supervision [11,12,13]. The less active pwCF showed 51 min/day of moderate-to-vigorous PA compared to the more active pwCF with 137 min/day of moderate-to-vigorous PA. Similar results were obtained in healthy children during the WHO Study of Health Behaviour in School-aged Children in Germany where only 10% of girls and 17% of boys fulfilled the recommendation of 60 min of moderate-to-vigorous physical activity. Regarding exercise training, 32% of the girls and 50% of the boys were doing sports 4 times per week [14]. In our cohort, only 43 of 88 pwCF met the given recommendation regarding the amount of PA, although all individuals of our cohort were part of the interventional group of the sport program with higher expected motivation to increase PA than non-participants. In the less active group, workload (Wpeak) improved over the course (*p* < 0.05), which may have resulted from a higher motivation in this severely affected CF-group.

The present results are in line with similar studies by Hebestreit et al. and Kriemler et al. In both studies, supervision produced a long-lasting effect even after supervised exercise ended, and it declined 18 months after end of intervention and supervision [7,28]. It may be concluded that behavioral change can only be achieved by continuous supervision and counselling throughout an exercise program. Through continued supervision, patients may be motivated and barriers that might prevent participation in PA may be identified early. Thus, supervision needs to be maintained over a long period of time and, thus, should be routinely implemented in routine CF care.

In a recently published review, barriers and facilitators to PA among children, adolescents, and adults with CF were assessed by means of a semi-standardized interview [16]. All items found in the latest studies, for example, “good accessibility of sports facilities”, “lack of time”, and “tiredness” were also subject in the validated questionnaire of Krämer et al. that we used [20]. These items were not only addressed in cohorts of pwCF but also in patients with COPD and chronic back pain [16,17,18]. Similar to adolescents with age-appropriate development, parental support, family involvement, opportunities to exercise, new social networks, and access to programs and facilities, pwCF were also strongly associated with PA participation [16,29,30]. The other barriers to PA such as a “lack of time” and “lack of interest” were similar to the barriers in pwCF [16,29,30]. Our hypothesis that supervision has a beneficial effect on barrier management could not be confirmed in either the entire group or in the subgroups. 

Interestingly, physical barriers were shown to have a more significant impact on participation in sport and exercise than psychosocial barriers. This fact could be related to self-observation, evaluation, and reward skills, as well as strategies for dealing with failure. Participants and the sports therapist worked together to develop an exercise program that considered inclinations and interests, as well as demands of daily life. In addition, participants were provided with exercise equipment from a pool at the clinic to use over a period of time. It is possible that the expectations of the pwCF to achieve a rapid increase in physical performance through training with the equipment were not fulfilled. In addition, a discrepancy between the objective and subjective physical performance could play a role, leading to greater physical barriers due to subjective perceived failure.

This is especially important for sport and exercise for young people with CF, as a vulnerable group. Like healthy peers, many prefer to perform sporting activities outside of sports clubs, together with their friends, regardless of set times. Informal sports can be an approach to minimize barriers to participation and increase motivation to participate. In addition, the previously listed questions were considered and implemented here as an action plan.

Another reason could be the different seasons in which the pwCF were studied. For example, in the colder seasons, increased coughing or shortness of breath could play a greater role and present more significant physical barriers than in warmer seasons.

In the present study, the pwCF were encouraged to formulate their motivational intentions regarding their participation in sport and exercise. In contrast, the volitional intention aimed to encourage pwCF to create a plan to address the questions of what, where, when, with whom, and how they will engage in physical activity (implementation intentions). For long-term implementation in terms of regular participation of exercise and activities in everyday life, these 5 W-questions are important; they are the base on which to create an action plan. In doing so, the activities should have low-threshold barriers and be easy to implement without much financial and organizational effort.

Furthermore, answering the question of how means providing knowledge on how to perform physical activities. It must be conveyed that compliance with the recommendations for exercise is sufficient to maintain and also increase physical performance, although many of the children and young people as well as adult pwCF do not achieve these targets.

It is therefore very challenging for the sports therapists and CF care team to come up with practical ways for pwCF to increase daily exercise time. Digital media such as apps or digital active games could be a method to realize this not only for adolescents with CF. An exercise program by means of a digital app considering the five questions could be created together with the sport therapists. Another option could be walking at alternating high- and low-intensity intervals supported with music at different beats/minute. The barrier to implementation is very low, as this is easy to apply in everyday life (implementation intentions).

Our supervision considered two parallel lines of intervention: The first one aimed to reduce sedentariness (e.g., avoiding elevators, preferring stairs, etc), and the second one aimed to introduce a structured exercise program into daily life. However, our supervision over 6 months did not show a beneficial effect on the barriers and counter strategies of the participants. The focus of our study was sport counselling and motivational aspects of sports. In the Health Action Process Approach (HAPA), the motivational phase should be followed by a volitional phase. In the motivational phase, outcome expectation and risk perception together with a high self-efficacy lead to intentions such as “do more PA/sports”. Without building up coping strategies and expecting a high self-efficacy in maintaining physical exercise during the volitional phase, the intention to increase PA will not convert into action, and behavior modification will not be obtained [31]. We assume that we have underestimated the role of the volitional aspect as not all personal coping strategies were adequately recognized during the supervised exercise program. As a result, pwCF were unable to maintain these strategies after the end of supervision. Thus, a period of supervision of 6 months may be too short to create a change in behavior. It is worth mentioning in this context that close contact between the sports councilor and the participants has proven to be more successful than training without supervision and support [7,28].

No definitive conclusion can be drawn from our results as to which subgroup of pwCF could benefit most from supervision. However, we assume that adolescents could benefit a lot from a supervised PA program as it could be shown that motor performance-related fitness improved up to the age of 14 years old followed by a plateau or even a nonsignificant decrease after age 14 [32].

All of our participants were motivated and had the intention to “be more active”, but they needed precise action plans and concrete coping strategies to achieve the aim to “be more active” and to maintain such behavior. Furthermore, at all stages of the process, patients need targeted information to successfully manage the stage. According to HAPA, supervision should include aspects of the volitional phase, such as precise action plans and individual coping strategies for the risky situations. Regarding our participants, this may be more important to the less active group of pwCF.

The present study had several limitations. Although the CFmobil program was offered to all of our CF patients (n = 240), while 88 pwCF agreed to participate, only a total of 23 pwCF was a complete data set available, i.e., the questionnaire was filled in at all test time points. It is unclear whether the non-participants were already physically active or if they needed other support to increase their PA. The questionnaire was only handed to participants interested in the program, and we included no control group with healthy peers. For this reason, the results cannot be compared with healthy persons or non-supervised CF patients, which limits the significance of the results.

A possible bias was that our participants could already be highly motivated individuals, irrespective of the program offer. The answers of patients could have been influenced by bias or a view towards social acceptability. If this bias existed, it appears to have stayed the same over time.

## 5. Conclusions

In summary, data from this interventional study on the implementation of sports and daily activity by training assistance and supervision in CF patients show that CF patients with poorer lung function are less active than patients with better lung function, and the former encounter more physical barriers to engaging in sports. Secondly, physical barriers turned out to be more important than psychological barriers across the whole CF study cohort. In this regard, CF patients might differ from healthy subjects, and this points at the importance of deliberately addressing these disease-specific barriers to exercise. A slight (non-significant) increase in the exercise amount and intensity along with a significant increase in fitness could be achieved in the less active patient group. One of the future challenges is to implement disease-specific means to promote PA and sports in routine CF care require. Supervision and sport counselling in CF should also focus on volitional aspects, in addition to motivational aspects, and include individual development of acute situational and preventive counter strategies. Besides these aspects, the transfer of knowledge about living healthily and being physically active in daily life (frequency, intensity, time, and type of activity) is needed.

Thus, continuous physical activity counselling should be part of the CF treatment group and should meet every CF patient’s needs in routine care. In future studies, a larger number of pwCF should be observed regarding barriers to PA and counter strategies. In future exercise programs, good implementation skills should be part of the counselling, as they are urgently needed to keep the individual physically active in the long term. 

## Figures and Tables

**Figure 1 ijerph-19-13150-f001:**
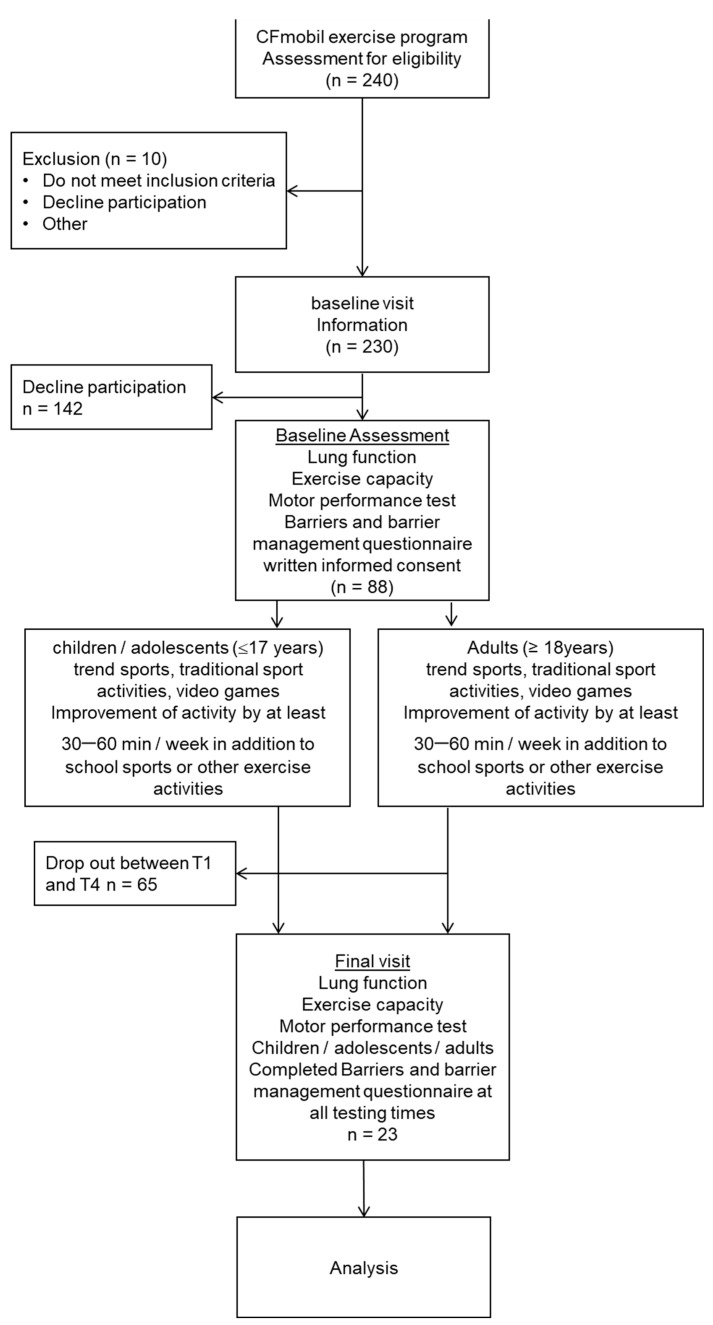
Selection flowchart of participants.

**Figure 2 ijerph-19-13150-f002:**
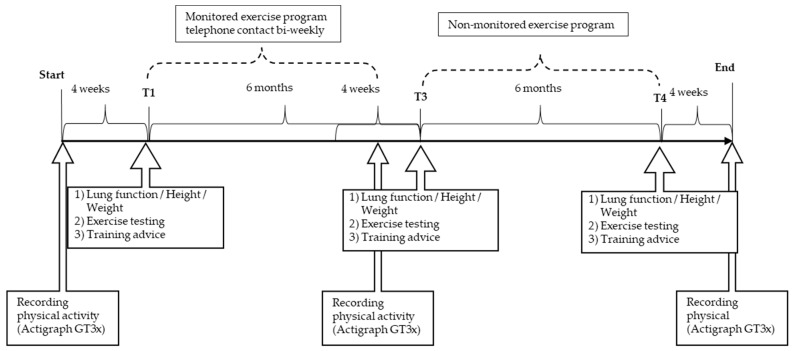
Timeline of CFmobil exercise program.

**Table 1 ijerph-19-13150-t001:** Anthropometric characteristics, lung function and steps/day, intensity of physical exercise, barriers, and counter strategies of the participants at baseline.

	Participants (n = 88)
	Mean ± SD
age (yrs)	24.2 ± 7.9
height (cm)	172.1 ± 10.0
weight (kg)	59.6 ± 12.4
BMI	20.0 ± 2.9
FEV1 (%pred)	56.3 ± 24.1
FVC (%pred)	72.7 ± 22.4
Wpeak (%pred)	63.8 ± 22.8
steps/day	8743 ± 3204
sedentary intensity (<1.5 METs) min/day	871 ± 189
light intensity (1.5–3 METs) min/day	480 ± 344
moderate to vigorous intensity (3–5.9 METs) min/day	103 ± 89
vigorous intensity (>6 METs) min/day	11 ± 13
situational psychosocial barriers (rating from 1–4)	1.95 ± 0.41
situational physical barriers (rating from 1–4)	2.43 ± 0.97
situational counter strategies (rating from 1–4)	2.43 ± 0.62
preventiv counter strategies (rating from 1–4)	2.37 ± 0.68

Abbreviations as described in text.

**Table 2 ijerph-19-13150-t002:** Anthropometric characteristics, lung function and steps/day, and intensity of physical exercise of participants by steps less than 7500 and more than 7500 steps at baseline.

	Less Than 7500 Steps (n = 45)	More Than 7500 Steps (n = 43)	
	Mean ± SD	Mean ± SD	*p*-Value
age (yrs)	24.2 ± 6.1	24.2 ± 9.0	0.741
height (cm)	169.6 ± 11.0	168.5 ± 10.3	0.298
weight (kg)	55.0 ± 9.5	58.9 ± 13.7 *	0.037
BMI	19.0 ± 23	20.5 ± 3.0 ***	0.000
FEV1 (%pred)	42.0 ± 17.8	68.2 ± 24.3 ***	0.000
FVC (%pred)	59.5 ± 22.1	83.1 ± 18.2 **	0.002
Wpeak (%pred)	52.0 ± 14.6	78.6 ± 21.0 ***	0.000
steps/day	5555 ± 1241	10786 ± 2254 ***	0.000
sedentary intensity (<1.5 METs) min/day	955 ± 206	810 ± 151 ***	0.000
light intensity (1.5–3 METs) min/day	350 ± 306	345 ± 202	0.089
moderate to vigorous intensity (3–5.9 METs) min/day	51 ± 36	137 ± 97 ***	0.000
vigorous intensity (>6 METs) min/day	7 ± 5	14 ± 13	0.337
situational psychosocial barriers (rating from 1–4)	1.94 ± 0.43	1.97 ± 0.42	0.890
situational physical barriers (rating from 1–4)	2.57 ± 1.06	2.40 ± 0.93	0.527
situational counter strategies (rating from 1–4)	2.37 ± 0.58	2.44 ± 0.67	0.801
preventative counter strategies (rating from 1–4)	2.31 ± 0.66	2.42 ± 0.68	0.450

Abbreviation as described in text, between groups according to the Mann-Whitney Test * *p* ≤ 0.05, ** = *p* < 0.01, *** = *p* < 0.001.

**Table 3 ijerph-19-13150-t003:** Anthropometric characteristics, lung function and steps/day, and intensity of physical exercise of participants by age.

	Age < 18 years (n = 15)	Age ≥ 18 years (n = 68)	
	Mean ± SD	Mean ± SD	*p*-Value
age (yrs)	14.8 ± 2.3	25.8 ± 7.4 ***	<0.001
height (cm)	164.6 ± 14.6	169.7 ± 9.4	0.338
weight (kg)	53.7 ± 16.9	57.9 ± 11.1	0.416
BMI	19.2 ± 3.2	20.0 ± 2.7	0.400
FEV1 (%pred)	81.5 ± 29.1	52.6 ± 21.0 **	0.007
FVC (%pred)	85.8 ± 24.8	70.8 ± 21.0	0.100
Wpeak (%pred)	80.6 ± 20.9	65.9 ± 22.1 *	0.047
steps/day	9937.0 ± 1956.0	8548.0 ± 3337.0	0.173
sedentary intensity (<1.5 METs) min/day	800.0 ± 152.0	882.0 ± 1193.0	0.246
light intensity (1.5–3 METs) min/day	327.0 ± 204.0	466.0 ± 348.0	0.173
moderate to vigorous intensity (3–5.9 METs) min/day	147.0 ± 119.0	96.0.0 ± 83.0	0.111
vigorous intensity (>6 METs) min/day	10.0 ± 8.0	11.0 ± 10.0	0.536
situational psychosocial barriers	1.97 ± 0.49	1.95 ± 0.40	0.834
situational physical barriers	1.96 ± 0.82	2.51 ± 0.97	0.132
situational counter strategies	2.59 ± 0.59	2.40 ± 0.63	0.645
preventative counter strategies	2.30 ± 0.70	2.39 ± 0.68	0.698

Abbreviations as described in text, between groups according to the Mann-Whitney-Test * *p* = <.0.05, ** = *p* < 0.01, *** = *p* < 0.001.

**Table 4 ijerph-19-13150-t004:** Anthropometric characteristics, lung function and steps/day, and intensity of physical exercise of participants by FEV1 < 40%pred./≥ 40%pred.

	FEV1 < 40% Pred. (n = 26)	FEV1 ≥ 40% Pred. (n = 62)	
	Mean ± SD	Mean ± SD	*p*-Value
age (yrs)	24.6 ± 5.5	23.8 ± 8.6	0.357
height (cm)	165.8 ± 8.4	170.0 ± 11.1	0.161
weight (kg)	52.7 ± 8.9	58.7 ± 13.0	0.57
BMI	19.1 ± 2.6	20.1 ± 2.9	0.219
FEV1 (%pred)	30.0 ± 7.5	68.0 ± 20.7 ***	<0.001
FVC (%pred)	47.3 ± 26.0	83.2 ± 16.2 ***	<0.001
Wpeak (%pred)	49.1 ± 16.0	73.1 ± 19.8 ***	<0.001
steps/day	6324.0 ± 2589	9384.0 ± 2829.0 ***	<0.001
sedentary intensity (<1.5 METs) min/day	975.0 ± 171.0	863.0 ± 177.0	0.091
light intensity (1.5-3 METs) min/day	468.0 ± 402	495.0 ± 338.0	0.252
moderate to vigorous intensity (3–5.9 METs) min/day	82.0 ± 74.0	106.0 ± 83.0	0.068
vigorous intensity (>6 METs) min/day	7.0 ± 7.0	13.0 ± 11.0	0.337
situational psychosocial barriers	1.89 ± 0.32	1.94 ± 0.44	0.780
situational physical barriers	2.38 ± 1.02	2.45 ± 0.97	0.770
situational counter strategies	2.38 ± 0.47	2.44 ± 0.68	0.528
preventative counter strategies	2.17 ± 0.60	2.44 ± 0.58	0.143

Abbreviations as described in text, between groups according to the Mann-Whitney-Test; *** = *p* < 0.001.

**Table 5 ijerph-19-13150-t005:** Anthropometric characteristics, lung function and steps/day, and intensity of physical exercise of participants by less than 7500 steps and more than 7500 steps during the monitored exercise program CFmobil.

	Less Than 7500 Steps (n = 12)			More Than 7500 Steps (n = 11)				
	Mean ± SD T1	Mean ± SD T3	Mean ± SD T4	Mean ± SD T1	Mean ± SD T3	Mean ± SD T4	Chi-Quadrat	*p*-Value
age (yrs)	25.3 ± 7.1			24.7 ± 9.9				
height (cm)	175.2 ± 9.5			168.5 ± 10.3				
weight (kg)	57.4 ± 8.0	57.9 ± 8.6	57.9 ± 8.1	59.6 ± 14.8	61.5 ± 16.9	62.7 ± 15.7	4.651	0.098
BMI	18.7 ± 1.5	18.8 ± 1.9	18.8 ± 1.8	20.7 ± 3.2	21.1 ± 3.9 #	21.3 ± 3.6 #	3.402	0.182
FEV1 (%pred)	45.6 ± 19.6	43.1 ± 20.9	47.8 ± 22.4	64.5 ± 19.6	69.0 ± 25.4	63.5 ± 23.9	7.304	0.026
FVC (%pred)	60.4 ± 18.2	58.4 ± 22.0	61.1 ± 21.0	86.2 ± 21.3 #	82.8 ± 20.2 #*	80.1 ± 20.2 #*	7.828	0.020
Wpeak (%pred)	53.0 ± 15.9	53.2 ± 18.4	59.1 ± 13.6 *	74.8 ± 21.3 #	76.6 ± 27.4 #	77.8 ± 18.9 #	7.929	0.019
steps/day	5650 ± 1565	6113 ± 2954	6263 ± 2905	10,272 ± 1828 ###	10,567 ± 3583 ##	10,098 ± 2883 #	0.333	0.846
sedentary intensity (< 1.5 METs) min/day	955 ± 206	820 ± 198 *	851 ± 140 *	810 ± 151 #	809 ± 175	754 ± 214 *	10.167	0.006
light intensity (1.5–3 METs) min/day	350 ± 306	255 ± 79	265 ± 41	564 ± 345 #	485 ± 284 #	338 ± 100 #	0.254	0.881
moderate to vigorous intensity (3–5.9 METs) min/d	47 ± 29	51 ± 36	41 ± 28	94 ± 66 #	137 ± 97 #	105 ± 62 #	1.000	0.607
vigorous intensity (>6 METs) min/day	7 ± 5	12 ± 11	10 ± 8	14 ± 15	14 ± 13	15 ± 14	1.811	0.404
situational psychosocial barriers (rating from 1–4)	1.98 ± 0.46	1.96 ± 0.42	1.93 ± 0.41	1.86 ± 0.46	1.68 ± 0.92	1.80 ± 0.44	1.455	0.453
situational physical barriers (rating from 1–4)	2.75 ± 1.10	2.88 ± 0.78	2.47 ± 1.12	2.29 ± 1.07	2.21 ± 0.92	2.45 ± 1.04	0.107	0.948
situational counter strategies (rating from 1–4)	2.56 ± 0.74	2.70 ± 0.68	2.63 ± 0.56	2.28 ± 0.72	2.22 ± 0.60	2.26 ± 0.52	0.325	0.850
preventative counter strategies (rating from 1–4)	2.53 ± 0.55	2.57 ± 0.76	2.46 ± 0.66	2.33 ± 0.54	2.07 ± 0.54	2.42 ± 0.67	0.481	0.788

Abbreviations as described in text; T1 = baseline, T3 = after 6 months, T4 = end of program (12 months); Friedman two-way Analysis of Variance (ANOVA) by Ranks Test; within groups Wilcoxon Test * *p*= <.0.05, between groups according to the Mann–Whitney U Test # ≤.0.05, ## = *p* < 0.01, ### = *p* < 0.001.

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
