# Peer review of "Barriers for Sports and Exercise Participation and Corresponding Barrier Management in Cystic Fibrosis"

_ijerph, 2022, doi:10.3390/ijerph192013150_

Round 1
Reviewer 1 Report (Previous Reviewer 1)
I thank the authors for their review and I think the paper is now suitable for publication
Reviewer 2 Report (Previous Reviewer 2)
I agree to publish this manucript.
This manuscript is a resubmission of an earlier submission. The following is a list of the peer review reports and author responses from that submission.
Round 1
Reviewer 1 Report
This paper describes a study aimed to assess both physical and psycho-social barriers for performing physical activity and their management, and to explore if supervision during an exercise program might have a beneficial effect.
88 subjects completed a questionnaire that assessed barriers, but only 23 completed the same at the end of the study period.
Authors' conclusion is that physical barriers seemed to have a higher priority not to participate in PA / exercise and supervision over 6 months during an exercise program did not show a beneficial effect on barriers and barrier management.
The study may be interesting for CF specialists, as physical activity for CF patients is essential to keep them in good health and adherence to physical activity may be lacking.
I hope my comments can help improve this paper

Reviewer 2 Report
The aim of the study was to assess the barriers for PA and the barrier management and explore the effect of supervision to the barriers and barrier management during an exercise program. Though some results were inconsistent with expectations, there were also some new and interesting findings. For example, the results showed that continuous supervision is important for early detection of barriers that prevent patients from engaging in physical activity and should be incorporated into the patient's daily behavior strategy. However, I have some questions talked about with authors.
1. The result that physical barriers turned out to be more important barriers than psychological barriers across the whole CF study cohort seemed to contradict three main factors including propensity factors, enablers and reinforcers that generally affected behavior. Would you explain it?
2. The result that besides the motivational aspect of sport counselling the volitional aspect seemed to be more important to incorporate more PA into daily life. Do you have any good suggestions and opinions?
Please further discuss and analyze the above two questions in depth.
Reviewer 3 Report
I reviewed the author's studies on cystic fibrosis before the publication of this paper and I would like to give this paper a high evaluation in that it is the result of a long study
